# TRIM28 SUMOylates and stabilizes NLRP3 to facilitate inflammasome activation

Ying Qin[1], Qi Li[1], Wenbo Liang[1], Rongzhen Yan[1], Li Tong[1], Mutian Jia ⬤ [1], Chunyuan Zhao[1,2] & Wei Zhao ⬤ [1✉]

The cellular NLRP3 protein level is crucial for assembly and activation of the NLRP3 inflammasome. Various posttranslational modifications (PTMs), including phosphorylation and ubiquitination, control NLRP3 protein degradation and inflammasome activation; however, the function of small ubiquitin-like modifier (SUMO) modification (called SUMOylation) in controlling NLRP3 stability and subsequent inflammasome activation is unclear. Here, we show that the E3 SUMO ligase tripartite motif-containing protein 28 (TRIM28) is an enhancer of NLRP3 inflammasome activation by facilitating NLRP3 expression. TRIM28 binds NLRP3, promotes SUMO1, SUMO2 and SUMO3 modification of NLRP3, and thereby inhibits NLRP3 ubiquitination and proteasomal degradation. Concordantly, *Trim28* deficiency attenuates NLRP3 inflammasome activation both in vitro and in vivo. These data identify a mechanism by which SUMOylation controls the cellular NLRP3 level and inflammasome activation, and reveal correlations and interactions of NLRP3 SUMOylation and ubiquitination during inflammasome activation.

---

[1] Department of Pathogenic Biology and Key Laboratory of Infection and Immunity of Shandong Province, School of Basic Medical Science, Cheeloo College of Medicine, Shandong University, Jinan, Shandong, China. [2] Department of Cell Biology, School of Basic Medical Science, Cheeloo College of Medicine, Shandong University, Jinan, Shandong, China. ✉email: wzhao@sdu.edu.cn

NOD-, LRR-, and pyrin domain-containing protein 3 (NLRP3) is a crucial pattern recognition receptor, which can sense pathogen-associated molecular patterns (PAMPs) and endogenous danger signals (danger-associated molecular patterns, DAMPs) to initiate the formation of a multi-protein complex named the NLRP3 inflammasome[1–5]. Canonical activation of the NLRP3 inflammasome requires two sequential steps. PAMPs or the activation of cytokines provides the first signal (priming signal), which activates nuclear factor (NF)-κB and induces the expression of the NLRP3, an integral part of the NLRP3 inflammasome, pro-interleukin-1β (pro-IL-1β), and pro-IL-18. The second signal (activation signal) is provided by a wide range of stimuli, such as particulates, crystals, and adenosine 5'-triphosphate disodium (ATP). Upon activation, NLRP3 inflammasome assembly activates caspase-1, which in turn cleaves pro-IL-1β and pro-IL-18. NLRP3 inflammasome has been implicated in multiple diseases and its activity should be tightly controlled[1].

Various posttranslational modifications (PTMs) of NLRP3, including phosphorylation, ubiquitination, and SUMOylation, play crucial roles in controlling inflammasome activation[1]. The level of intracellular NLRP3 protein is considered as a key rate-limiting step for canonical activation of NLRP3 inflammasome[6,7]. In resting macrophages, NLRP3 is ubiquitylated and its protein level is relatively low, which limits NLRP3 inflammasome activation. Upon priming and activation, NLRP3 is deubiquitylated, and that enhances NLRP3 expression and facilitates inflammasome activation[8,9]. The E3 ubiquitin ligases tripartite motif containing protein 31 (TRIM31) and membrane-associated RING finger protein 7 (MARCH7) induce Lys(K)48-linked ubiquitylation of NLRP3 and in turn promotes its degradation via proteasome and autophagy, respectively[10,11]. The ubiquitin-specific peptidase 1 (USP1)-associated factor 1 (UAF1) /USP1 deubiquitinase complex selectively inhibits K48-linked ubiquitination and protein degradation of NLRP3 enhances cellular NLRP3 levels[12], which are indispensable for subsequent NLRP3 inflammasome assembly and activation. Recently, small ubiquitin-like modifier (SUMO) protein-conjugated modifications, which are termed as

SUMOylation, were identified in NLRP3 during inflammasome activation[13,14]. SUMOylation, a type of PTM that is similar to ubiquitination, is mediated by E1-activating enzymes, a sole E2 ligase UBC9, and E3 SUMO ligases[15–17]. Three SUMO proteins, including SUMO1, SUMO2, and SUMO3, can be covalently conjugated to proteins as a single moiety (SUMO1) or as polymeric SUMO chains (SUMO2 and SUMO3). SUMOylation contributes significantly to mediating protein interaction, regulating protein localization in cells, regulating the activity of transcription factors, maintaining genomic stability, and transcriptional regulation[15]. The E3 SUMO protein ligase MUL1 (also known as MAPL) catalyzes SUMO2/3 modification of NLRP3 and restrains NLRP3 activation, whereas SUMO-1-catalyzed SUMOylation of NLRP3 facilitates inflammasome activation. Although SUMO1 modification of NLRP3 is required for inflammasome activation, both the mechanism details and the E3 SUMO ligase that mediates SUMO1-catalyzed SUMOylation of NLRP3 are unclear.

Here we identify the E3 SUMO ligase TRIM28 (also known as Kruppel-associated box-associated protein 1 (KAP1) or transcriptional intermediary factor 1β (TIF1β)) as an enhancer of inflammasome activity by promoting NLRP3 SUMOylation. TRIM28 interacts with NLRP3, facilitates SUMO1 and SUMO2/3 catalyzed SUMOylation of NLRP3, and thereby inhibits its ubiquitination and proteasomal degradation. TRIM28 stabilizes NLRP3 protein expression and enhances subsequent inflammasome activation. These data provide a mechanism by which SUMOylation enhances NLRP3 inflammasome activation and indicate that TRIM28 might be a target for the treatment of diseases caused by aberrant inflammasome activation.

## Results

**TRIM28 interacts with NLRP3.** To identify potential interactors of NLRP3, we incubated a specific anti-NLRP3 antibody with total cell lysates of mouse primary peritoneal macrophages (PMs) stimulated with lipopolysaccharide (LPS). The co-immunoprecipitated complex was separated by sodium dodecyl sulfate–polyacrylamide gel electrophoresis (SDS–PAGE). The E3 SUMO ligase TRIM28 was

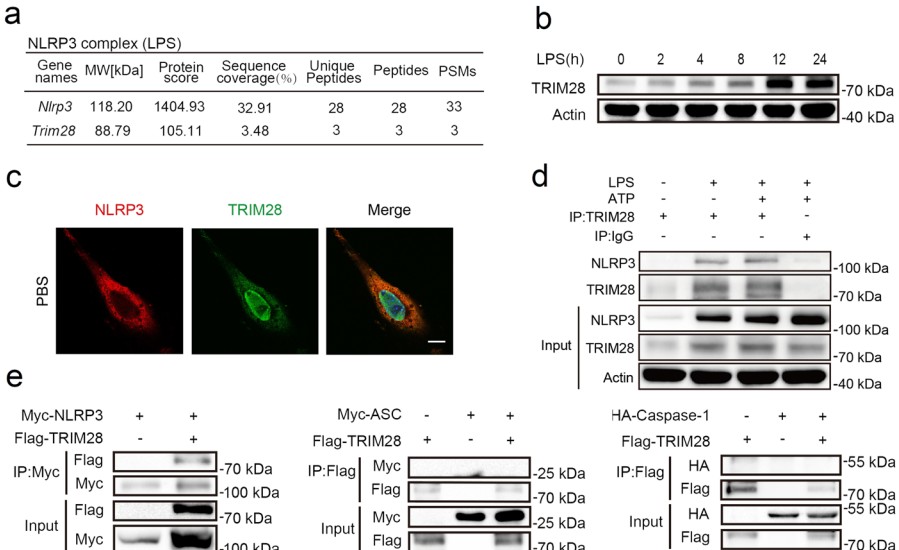

**Fig. 1 TRIM28 interacts with NLRP3. a** Identification of TRIM28 as a potential NLRP3 interactor in LPS-stimulated mouse peritoneal macrophages (PMs) by mass spectrum primed with LPS. **b** Immunoblot analysis of TRIM28 expression in LPS-stimulated mouse PMs (n = 3 independent experiments). **c** Confocal microscopic analysis of the colocalization of NLRP3 (Red) and TRIM28 (Green) in unstimulated mouse embryonic fibroblasts (MEFs). Scale bar, 10 μm. **d** IP analysis of endogenous association between TRIM28 and NLRP3 in LPS-stimulated or LPS-primed and ATP-activated mouse PMs (n = 3 independent experiments). **e** Immunoprecipitation (IP) analysis of the association between TRIM28 and NLRP3, ASC or Caspase-1 in HEK293T cells transfected with the indicated plasmids. Similar results were obtained from three independent experiments.

identified as a potential NLRP3 interactor by liquid chromatography coupled with tandem mass spectrometry (Fig. 1a). LPS markedly induced TRIM28 expression (Fig. 1b), suggesting that TRIM28 may be involved in the process of NLRP3 inflammasome activation. However, *Trim28* mRNA decreased at first and increased subsequently upon LPS stimulation (Supplementary Fig. 1a), indicating the existence of other regulatory mechanisms for TRIM28 protein levels in response to LPS stimulation. We next examined the interaction between TRIM28 and NLRP3 in vivo. Confocal analysis demonstrated the colocalization between TRIM28 and NLRP3 (Fig. 1c). Next, the interaction between TRIM28 and NLRP3 was investigated under physiological conditions by immunoprecipitation (IP). An association between TRIM28 and NLRP3 was detected in both resting LPS-primed and ATP-activated mouse PMs (Fig. 1d). TRIM28 along with NLRP3, ASC, or caspase-1 were cotransfected in HEK293T cells. TRIM28 could immunoprecipitate with NLRP3 but not ASC and caspase-1 (Fig. 1e). Taken together, these data suggest that TRIM28 could interact with NLRP3.

**TRIM28 has no effects on NLRP3 inflammasome priming**. Next, we first investigated the potential functions of TRIM28 in the priming of NLRP3 inflammasome. *Trim28*fl/fl mice were crossed with *Lyz2*cre mice to specifically knockout *Trim28* in myeloid cells (called "*Trim28*CKO" here). *Trim28* deficiency had no effects on LPS-induced phosphorylation of IκBα and total IκBα (Fig. 2a) and transcription of *Il1b*, *Nlrp3*, *Tnf*, and *Il6* in PMs (Fig. 2b and Supplementary Fig. 1b). To further investigate the function of TRIM28, small interfering RNA (siRNA)-knockdown experiments were performed. Synthesized siRNA targeting mouse *Trim28* was used to suppress endogenous TRIM28 expression in PMs (Supplementary Fig. 1c). *Trim28* knockdown also did not affect phosphorylation of IκBα and total IκBα expression and transcription of *Il1b*, *Nlrp3*, *Tnf*, and *Il6* (Fig. 2c, d and Supplementary Fig. 1d). Collectively, these data indicate that TRIM28 could not affect LPS-induced NF-κB activation and the priming process of NLRP3 inflammasome activation.

**TRIM28 promotes NLRP3 inflammasome activation**. Next, we tested the effect of TRIM28 on NLRP3 inflammasome activation. Mouse PMs from both wild-type (WT) and *Trim28*CKO mice were primed by LPS and then stimulated by NLRP3 inflammasome activators (ATP or nigericin (Nig)) or poly(dA:dT) (an AIM2 inflammasome activator). *Trim28* deficiency markedly inhibited NLRP3 inflammasome-induced IL-1β secretion, while had no effects on AIM2 inflammasome activation (Fig. 3a). However, tumor necrosis factor (TNF) and IL-6 secretion were not influenced by *Trim28* deficiency (Fig. 3b). Activated caspase-1 cleaves gasdermin D (GSDMD) and releases its N-terminal regions, which then punch holes in the membrane, resulting in the release of IL-1β and pyroptosis[18]. *Trim28* deficiency suppressed both caspase-1 and GSDMD cleavage following NLRP3 inflammasome activation (Fig. 3c). Meanwhile, we observed that *Trim28* deficiency did not influence IL-1β but greatly attenuated LPS-induced NLRP3 expression (Fig. 3c), suggesting that TRIM28 may selectively enhance NLRP3 inflammasome activation by promoting NLRP3 expression. *Trim28* knockdown also reduced IL-1β secretion following LPS priming and ATP activation in mouse PMs (Fig. 3d), with no effects on both TNF and IL-6 secretion (Supplementary Fig. 1e). Taken together, these data indicate that TRIM28 facilitates NLRP3 inflammasome activation.

We next investigated the function of TRIM28 on NLRP3 inflammasome activation in vivo. The secretion of IL-1β by intraperitoneal (i.p.) injection of LPS is NLRP3 dependent. The secretion of IL-1β in the serum of *Trim28*CKO mice decreased upon LPS challenge, while TNF and IL-6 remained unchanged (Fig. 3e). These results demonstrate that *Trim28* deficiency attenuates NLRP3 inflammasome activation in vivo.

**TRIM28 inhibits proteasomal degradation of NLRP3**. Next, we determined whether TRIM28 regulates NLRP3 protein expression. TRIM28 overexpression increased NLRP3 protein expression in HEK293T cell (Fig. 4a). Consistently, both *Trim28* deficiency and knockdown inhibited LPS-induced NLRP3 protein

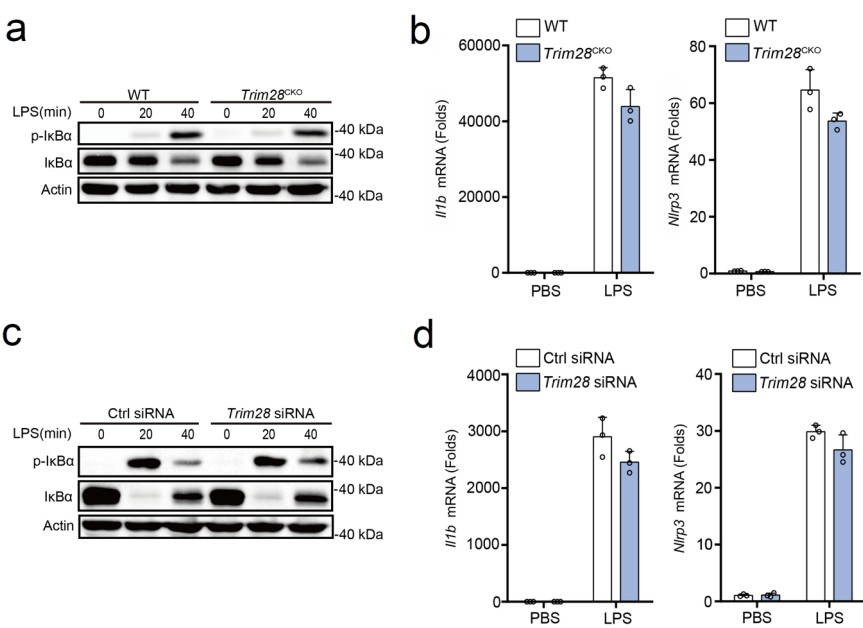

**Fig. 2 No effects of TRIM28 on NLRP3 inflammasome priming. a, c** Immunoblot analysis of p-IκBα and IκBα in LPS-stimulated mouse PMs from WT or *Trim28*CKO mice (**a**) or PMs transfected with Ctrl siRNA or *Trim28* siRNA (**c**). **b, d** RT-PCR analysis of *Ilb* and *Nlrp3* mRNA in LPS-stimulated mouse PMs from WT or *Trim28*CKO mice (**b**) or PMs transfected with Ctrl siRNA or *Trim28* siRNA (**d**). All data are represented as mean ± SD in **b**, **d**. Similar results were obtained from five independent experiments (**a**, **c**) and three independent experiments (**b**, **d**).

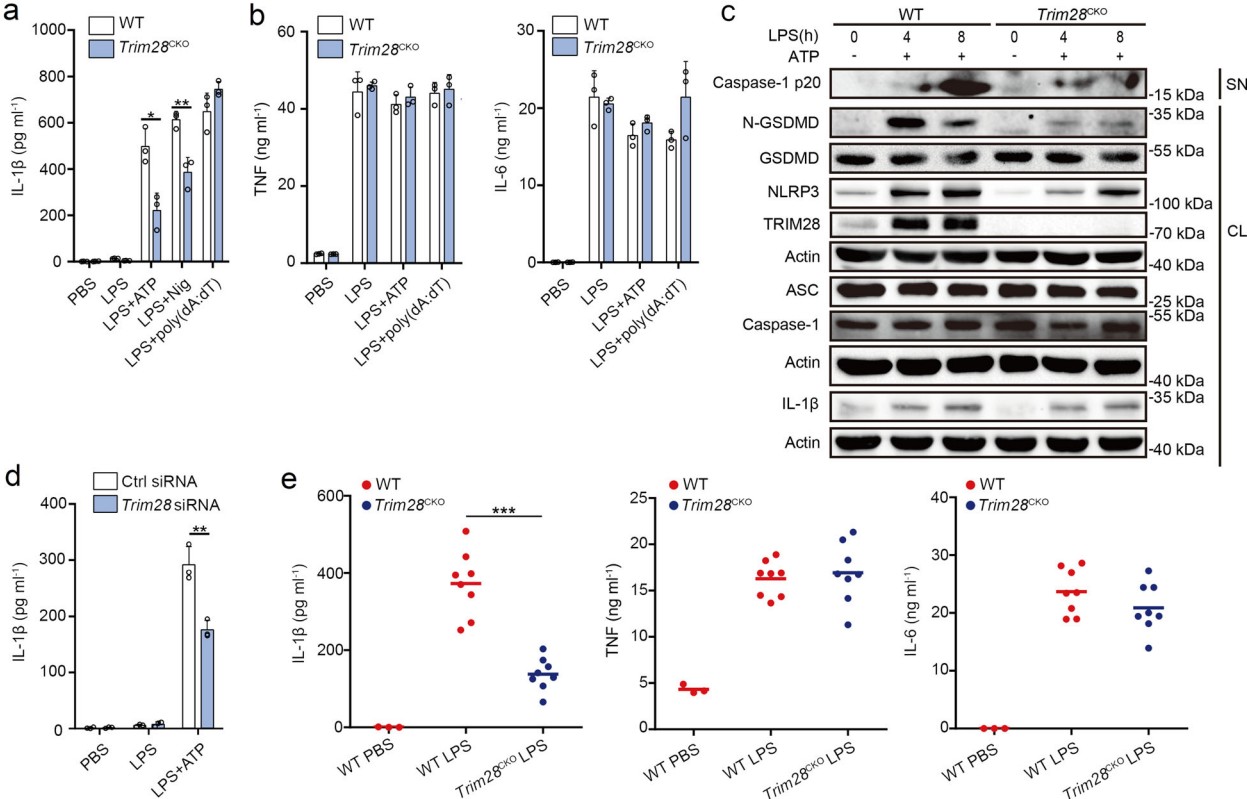

**Fig. 3 TRIM28 promotes NLRP3 inflammasome activation. a**, **b** ELISA analysis of IL-1β, TNF, and IL-6 in supernatants of mouse PMs from WT or Trim28CKO mice following priming with LPS for 7 h and subsequent stimulation with ATP, Nig, or poly(dA:dT) for 1 h (**a** two-tailed t test WT vs. Trim28CKO, \*$p = 0.011241$, \*\*$p = 0.005688$). **c** Immunoblot analysis of supernatants (SN) and cell lysates (CL) of mouse PMs from WT or Trim28CKO mice, following priming with LPS and subsequent stimulation with ATP ($n = 3$ independent experiments). **d** ELISA analysis of IL-1β in supernatants from mouse PMs transfected with control (Ctrl) siRNA or Trim28 siRNA for 48 h, followed by priming with LPS for 7 h and subsequent stimulation with ATP for 1 h (two-tailed t test Ctrl siRNA vs. Trim28 siRNA, \*\*$p = 0.005134$). **e** ELISA analysis of serum levels of IL-1β, TNF, and IL-6 of WT or Trim28CKO mice after i.p. LPS injection (PBS, $n = 3$; LPS, $n = 8$ per group, two-tailed t test WT vs. Trim28CKO, \*\*\*$p = 6 \times 10^{-6}$). All data are represented as mean ± SD in **a**, **b**, **d**, **e** (\*$p < 0.05$; \*\*$p < 0.01$; \*\*\*$p < 0.001$). Similar results were obtained from three independent experiments.

expression in mouse PMs (Fig. 4b, c). Interestingly, NLRP3 protein expression in unstimulated macrophages was also considerably attenuated by Trim28 deficiency or knockdown (Supplementary Fig. 2a, b). Although TRIM28 has transcriptional regulatory functions[19], it did not enhance NLRP3 transcription (Fig. 2b, d). We then investigated the effects of TRIM28 on NLRP3 protein degradation by cycloheximide (CHX) chase assay. Trim28 deficiency markedly promoted NLRP3 protein degradation (Fig. 4d, e). Ubiquitin–proteasome and autophagy–lysosome are major pathways for NLRP3 degradation. We then further clarified which pathway was affected by TRIM28. MG132, a selective inhibitor of proteasome, but not the lysosome inhibitor chloroquine and autophagy inhibitor 3-methyladenine (3-MA), restored the NLRP3 protein expression in LPS-stimulated Trim28 deficiency macrophages (Fig. 4f, g and Supplementary Fig. 2c). Collectively, these data indicate that TRIM28 selectively inhibits NLRP3 protein degradation in proteasome.

**TRIM28 promotes SUMOylation of NLRP3.** Ubiquitination is required for the protein degradation via the ubiquitin–proteasome pathway. We therefore examined the potential roles of TRIM28 on NLRP3 ubiquitination. Trim28 deficiency facilitated NLRP3 ubiquitination in unstimulated, LPS-primed, and ATP-activated macrophages (Fig. 5a). K48-linked ubiquitination is required for proteasomal degradation of target proteins. Next, to clarify whether TRIM28 could inhibit K48-linked ubiquitination, the ubiquitin mutant vectors K48 and K63, which contain arginine

substitutions of all of its lysine residues except the one at positions 48 and 63 respectively, were used in the transfection assays. TRIM28 inhibited both total and K48-linked ubiquitination of NLRP3, whereas it had no effects on K63-linked ubiquitination of NLRP3 (Fig. 5b). TRIM28 is an E3 SUMOylation ligase[20] and the cysteine (C) at position 651 (C651) is required for its enzymatic activity. TRIM28 point mutant (C651) with substitution of the cysteine residue with alanine (A) at position 651 could not bind to the key E2 SUMO binding enzyme UBC9 (Supplementary Fig. 3), and thus its E3 SUMO ligase activity was inactivated[21]. TRIM28 C651 lost the ability to inhibit NLRP3 ubiquitination and degradation (Fig. 5c, d), suggesting that TRIM28 selectively attenuates K48-linked ubiquitination of NLRP3 and its proteasomal degradation, via its E3 SUMO ligase activity.

NLRP3 could be modified by SUMO1 and SUMO2/3 in macrophages (Fig. 6a, b), which is consistent with previous reports[13,14]. Interestingly, we found that LPS stimulation enhanced both SUMO1 and SUMO2/3 modification of NLRP3, while SUMO1 modification further increased and SUMO2/3 modification decreased following ATP stimulation (Fig. 6a, b and Supplementary Fig. 4). These data suggest a dynamic change of NLRP3 SUMOylation status during the priming and activation step of NLRP inflammasome. 2′,3′,4′-Trihydroxy flavone (2-D08), an inhibitor of protein SUMOylation, dose-dependently decreased NLRP3 expression (Fig. 6c), suggesting the potential role of SUMOylation on NLRP3 expression. We have identified E3 SUMO ligase TRIM28 as a NLRP3-associated protein and that

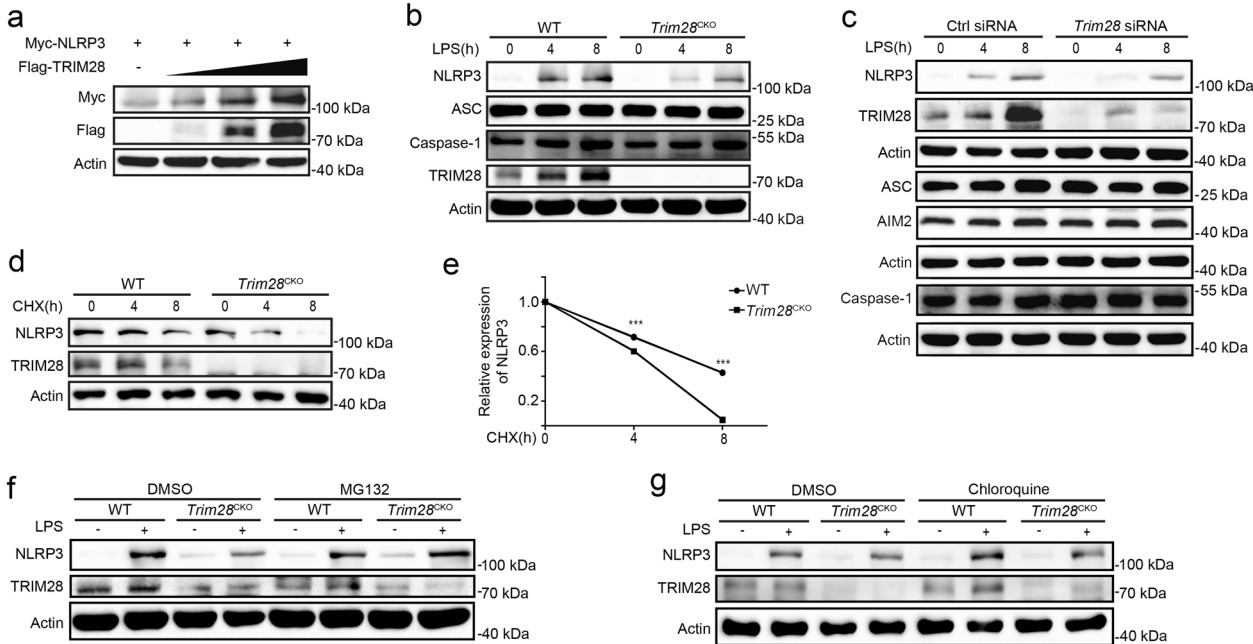

**Fig. 4 TRIM28 inhibits NLRP3 protein degradation. a** Immunoblot analysis of lysates from HEK293T cells transfected with Myc-NLRP3 and increasing amount of Flag-TRIM28 plasmid. **b** Immunoblot analysis of lysates of mouse PMs from WT or *Trim28*CKO mice, following stimulation with LPS. **c** Immunoblot analysis of lysates of mouse PMs transfected with Ctrl siRNA or *Trim28* siRNA for 48 h, following stimulation with LPS. **d**, **e** Immunoblot analysis of NLRP3 expression in mouse PMs from WT or *Trim28*CKO mice stimulated with LPS for 4 h and then treated cycloheximide (CHX) for the indicated time periods. NLRP3 expression was quantitated by measuring band intensities using the "ImageJ" software. The values were normalized to Actin. Data are represented as mean ± SD (two-tailed $t$ test WT vs. *Trim28*CKO, ***$p = 1 \times 10^{-12}$, $3 \times 10^{-17}$). **f** Immunoblot analysis of NLRP3 expression in mouse PMs from WT or *Trim28*CKO mice stimulated with LPS for 4 h, together with DMSO or MG132 (10 μM) treatment for 4 h. **g** Immunoblot analysis of NLRP3 expression in mouse PMs from WT or *Trim28*CKO mice stimulated with LPS for 4 h, together with DMSO or chloroquine (50 μM) treatment for 4 h. Similar results were obtained from three independent experiments.

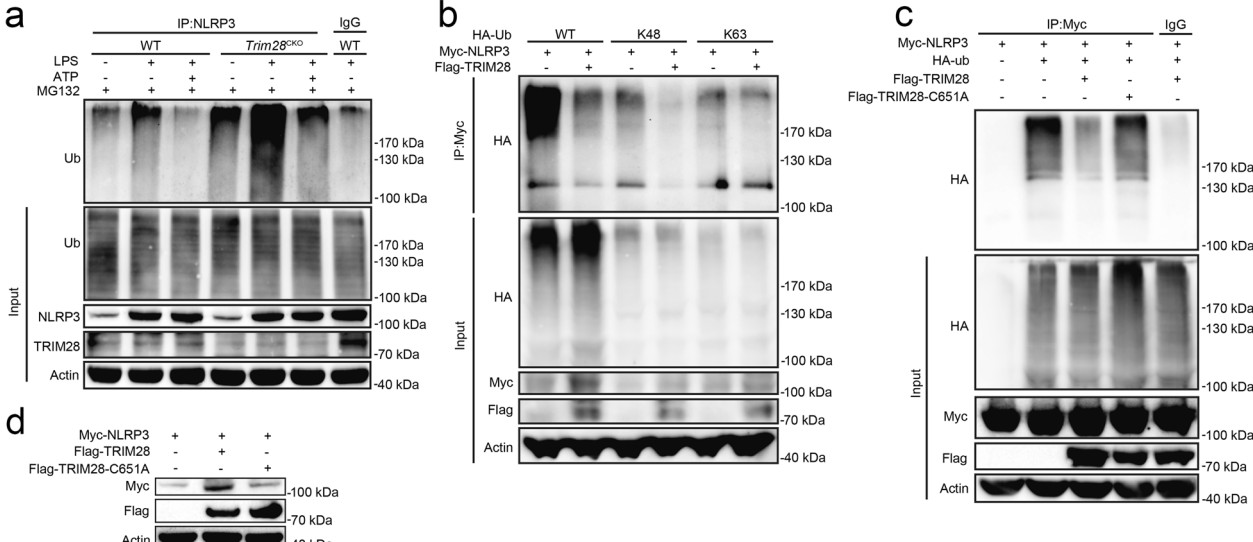

**Fig. 5 TRIM28 inhibits K48-linked ubiquitination of NLRP3. a** Immunoblot analysis of lysates of mouse PMs from WT or *Trim28*CKO mice, primed with LPS for 4 h, together with MG132 (10 μM) treatment for 3 h, and then stimulation with ATP for 1 h, and followed by IP with NLRP3 antibody. **b** Immunoblot analysis of lysates from HEK293T cells transfected with HA-Ub (WT), HA-tagged K48-linked ubiquitin (K48-Ub), or HA-tagged K63-linked ubiquitin (K63-Ub), Myc-NLRP3, and Flag-TRIM28, together with MG132 (10 μM) treatment for 4 h and followed by IP with Myc antibody. **c** Immunoblot analysis of lysates from HEK293T cells transfected with HA-tagged ubiquitin (HA-Ub), Myc-NLRP3, and Flag-TRIM28 WT or C651A, together with MG132 (10 μM) treatment for 4 h, and followed by IP with Myc antibody. **d** Immunoblot analysis of lysates from HEK293T cells transfected with Myc-NLRP3, together with Flag-TRIM28 WT or C651A. Similar results were obtained from three independent experiments.

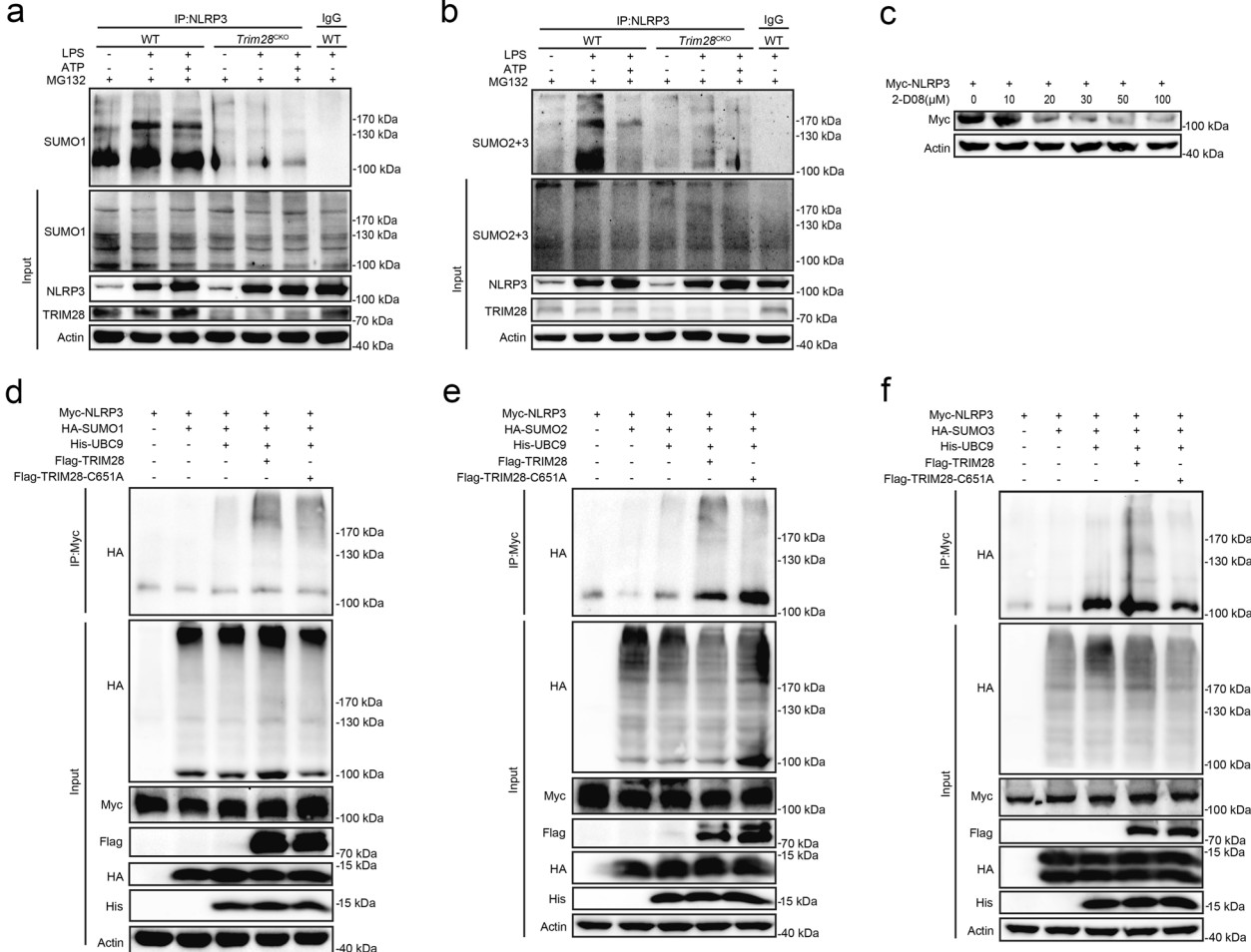

**Fig. 6 TRIM28 promotes SUMOylation of NLRP3. a, b** Immunoblot analysis of lysates of mouse peritoneal macrophages from WT or *Trim28*CKO mice, primed with LPS for 4 h, together with MG132 (10 μM) treatment for 3 h, and then stimulation with ATP for 1 h, and followed by SUMOylation assays with NLRP3 antibody. **c** Immunoblot analysis of lysates from HEK293T cells transfected with Myc-NLRP3 and stimulated with an increasing amount of 2-D08 for 24 h. **d–f** Immunoblot analysis of lysates from HEK293T cells transfected with Myc-NLRP3, His-UBC9 and Flag-TRIM28 WT or C651A, and HA-SUMO1, HA-SUMO2, or HA-SUMO3, followed by MG132 (10 μM) treatment for 4 h, and then performed SUMOylation assays with Myc antibody. Similar results were obtained from three independent experiments.

promoted us to investigate whether TRIM28 could mediate the SUMOylation of NLRP3. Both *Trim28* deficiency and knockdown attenuated SUMO1 and SUMO2/3 modification of NLRP3 in macrophages (Fig. 6a, b and Supplementary Fig. 4). Furthermore, TRIM28 promoted SUMO1, SUMO2, and SUMO3 modification in HEK293T cells, while TRIM28 C651 mutant lost the ability to mediate NLRP3 SUMOylation (Fig. 6d–f). Therefore, TRIM28 induced NLRP3 SUMOylation, inhibited NLRP3 ubiquitination and degradation, and thus facilitated inflammasome activation (see a model in Fig. 7).

## Discussion

Various types of PTMs are important for the regulation of protein folding, stability, localization, and functional activities. NLRP3 is subjected to multiple PTMs, including phosphorylation, ubiquitination, alkylation, S-nitrosylation, and SUMOylation, to tightly control optimal activation of NLRP3 inflammasome. But the correlations and interactions among various types of PTMs of NLRP3 remain less clarified. SUMOylation interacts with other PTMs, such as ubiquitination, phosphorylation, and acetylation, to regulate protein functions[15]. For example, ubiquitinated p53 recruits the E3 SUMO ligase PLASy to promote SUMOylation of p53[22]. E3 SUMO ligase PLAS4-mediated SUMOylation of MDC1

recruits E3 ubiquitin-ligase RNF4 to promote MDC ubiquitination in DNA damage[16]. However, SUMO1 modification usually blocks ubiquitination of its substrate. For example, SUMO1-modified IκBα cannot be ubiquitinated and is resistant to proteasome-mediated degradation[23]. SUMO1 modification of cGAS and STING prevent their polyubiquitination and degradation during viral infection[24]. SUMO1 and Ub both bind to the lysine (K) residues of their substrates and that leads to competitive inhibition. In this study, we reveal the correlations and interactions of NLRP3 SUMOylation and ubiquitination during inflammasome activation. The E3 SUMO ligase TRIM28 facilitates SUMO1 and SUMO2/3-catalyzed SUMOylation of NLRP3, whereby attenuates K48-linked ubiquitination of NLRP3, resulting in enhancement of NLRP3 stability.

During NLRP3 inflammasome priming (such as LPS stimulation), SUMO1/2/3 modification and ubiquitination are all increased. Following activation (ATP stimulation), SUMO1 modification is further enhanced, while SUMO2/3 modification and ubiquitination were decreased, suggesting that dynamic changes of SUMOylation and ubiquitination during NLRP3 inflammasome activation. MUL1-mediated SUMO2/3-catalyzed NLRP3 SUMOylation was reported to inhibit NLRP3 inflammasome activation[13]. But SUMO1-catalyzed NLRP3 SUMOylation promotes inflammasome activation, which is consistent with

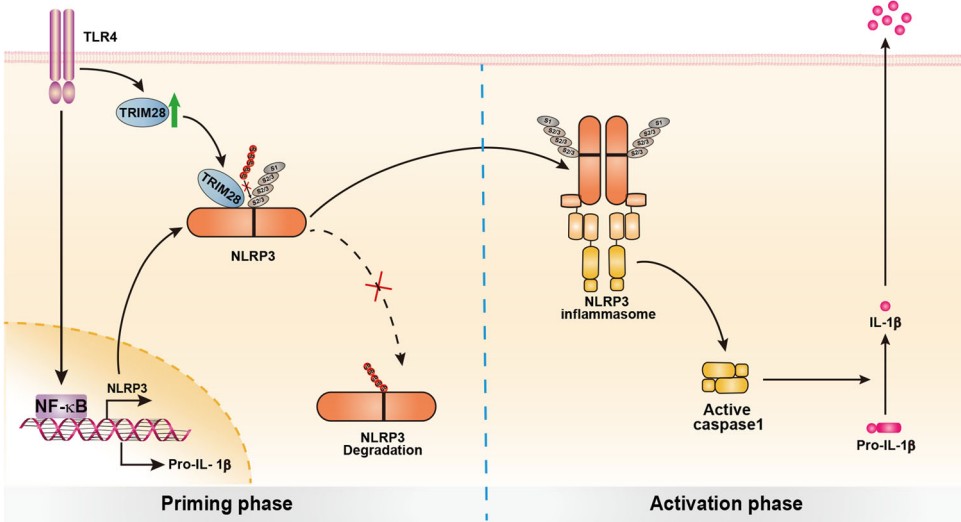

**Fig. 7 Working model for TRIM28 promoting NLRP3 inflammasome activation.** TRIM28 binds to NLRP3, promotes SUMOylation of NLRP3, and thus inhibits NLRP3 ubiquitination and degradation in proteasome, resulting in the enhancement of NLRP3 inflammasome activation.

a previous report[14]. SUMO1 modified Lys (K) residues cannot be SUMOylated or ubiquitinated, which is different to the characteristics of SUMO2/3 modification and ubiquitination[25,26]. Therefore, SUMO1 modification of NLRP3 may block the conjugation of ubiquitin by competing with the same K sites of NLRP3 and inhibits ubiquitous degradation of NLRP3 and therefore facilitates inflammasome activation. Although MUL1-mediated SUMO2/3-catalyzed SUMOylation of NLRP3 inhibited inflammasome activation[13], we found that TRIM28-mediated SUMO2/3-catalyzed SUMOylation facilitated inflammasome activation. In addition, TRIM28, but not MUL1, inhibited the ubiquitination degradation of NLRP3. Therefore, TRIM28 and MUL1 regulate NLRP3 activity by different mechanisms. Multiple potential SUMOylation sites exist in NLRP3 protein (such as K204 and K689, etc.), and TRIM28 and MUL1 may mediate NLRP3 SUMOylation at different lysine sites. SUMOylation mediated by TRIM28 may block the ubiquitination site of NLRP3 and thus inhibits NLRP3 proteasomal degradation. Further analysis of the SUMOylation and ubiquitination sites of NLRP3 will help to elucidate the underlying mechanisms. In addition, TRIM28 promotes SUMO1/2/3-catalyzed SUMOylation of NLRP3, and SUMO1 blocks other PTMs (including ubiquitination) of NLRP3, resulting in the stability of NLRP3 and enhancement of inflammasome activation. Based on the experimental data, we propose a working model to illustrate how TRIM28-mediated SUMOylation enhances NLRP3 inflammasome activation (Fig. 7). TRIM28 binds to NLRP3, promotes its SUMOylation, and thereby inhibits NLRP3 ubiquitination and subsequent degradation in proteasome. However, in Trim28-deficiency macrophages, NLRP3 is still synthesized normally, and NLRP3 degradation is enhanced. Therefore, the absence of TRIM28 does not completely eliminate NLRP3 expression. Thus, enhanced cellular level of NLRP3 facilitates inflammasome assembly and enhances inflammasome activation.

In addition to its E3 SUMO ligase activity, TRIM28 is a well-known epigenetic suppressor. TRIM28 can bind to KRAB zinc-finger proteins (KZNFs) and recruit the histone methyltransferase SETDB1 to recognize and bind to H3k9me3, resulting in transcriptional repression[27–29]. TRIM28 works as a master epigenetic regulator of endogenous retroviral elements (ERVs) to maintain ERVs in a silent state and thereby protects cells from detrimental genome instability and aberrant interferon responses. Furthermore, TRIM28 possesses important epigenetic roles in the

differentiation and functions of regulatory T cells and Th17 cells, and genetic deletion of KAP1 in T cells results in autoimmune phenotype[30–33]. However, in our study, we found that TRIM28 had no effect on the transcription of key components of inflammasome, including Il1b and Nlrp3.

In summary, we identify the E3 SUMO ligase TRIM28 as a stabilizer of NLRP3 by promoting its SUMOylation. Concordantly, TRIM28 enhances inflammasome activation via enhancing cellular level of NLRP3. Given the pivotal roles of inflammasome in the occurrence and development of a variety of diseases, modulation of NLRP3 activation by targeting SUMOylation or TRIM28 will be a priming target for the treatment of diseases caused by aberrant NLRP3 activity.

## Methods
**Mice.** Trim28[fl/fl] mice[34] (Stock Number:018552) and Lyz2[cre] mice (Stock Number: 004781) were from Jackson Laboratory. Trim28[fl/fl] mice were crossed with Lyz2[cre] transgenic mice to obtain Trim28[CKO] mice with Trim28 deficiency in myeloid cells. Mice were kept in specific pathogen-free-level Laboratory Animal Room with 12-/12-h light and dark alternating time, temperature 18–22 °C and humidity 45–65%. The background strain of all mice used is C57BL/6J. All mice were used at 6–11 weeks of age, and both male and female mice were used. All animal experiments were undertaken in accordance with the National Institutes of Health Guide for the Care and Use of Laboratory Animals, with the approval of the Scientific Investigation Board of the School of Basic Medical Science, Shandong University, Jinan, Shandong Province, China.

**Reagents and antibodies.** ATP (A1852), Nig (N7143), Z-Leu-Leu-Leu-al (MG132, C2211), LPS (Escherichia coli, O111:B4, L4130), N-ethylmaleimide (NEM, E3876), anti-Myc (M4439, 1:5000), anti-HA (H3663, 1:1000), and anti-Flag M2 (F1804, 1:1000) were from Sigma-Aldrich (St Louis, MO); chloroquine diphosphate (S4157), 3-MA (5142-23-4) and 2-D08 (S8696) were from Selleck; CHX (A8244) was from APExBIO Technology; poly(dA:dT) (tlrl-patn) was from Invivogen (San Diego, CA); anti-mouse IgG (7076), anti-p-IκBα (2859P, 1:1000), anti-IκBα (4814, 1:1000), anti-IL-1β (12242S, 1:1000), anti-AIM2 (#13095 S, 1:1000), anti-SUMO1 (4930T, 1:1000), and anti-SUMO-2/3 (18H8) (4971T, 1:1000) were from Cell Signaling Technology; anti-Caspase-1 p20 (AG-20B-0042, 1:1000), anti-NLRP3 (AG-20B-0014, 1:1000), and anti-ASC (AG-25B-0006, 1:1000) were from Adipo-Gen; anti-Caspase-1 (GTX14367, 1:1000) was from GenTex; protein G agarose (sc-2002) used for IP, anti-Ub (sc-8017, 1:1000) were from Santa Cruz Biotechnology (Santa Cruz, CA); anti-TRIM28 (ab22553, 1:1000) and anti-GSDMD (ab209845, 1:1000) were from Abcam (Cambridge, MA); anti-NLRP3 (19771-1-AP, 1:1000), anti-His (66005-1-Ig, 1:1000), horseradish peroxidase-conjugated Affinipure Goat Anti-Rabbit IgG (H+L) (SA00001-2, 1:5000), and anti-β-actin (66009-1-Ig, 1:2000) were from Proteintech; Alexa Fluor 633 (A-21071, 1:500) and Alexa Fluor 488 (A-11059, 1:500) were from Thermo Fisher Scientific; anti-Myc (N7143, 1:1000) was from ORIGENE. Anti-Flag and anti-Myc are the same isotypes.

**Cell culture**. To obtain mouse primary PMs, C57BL/6J mice (male, 6–8 weeks old) were injected i.p. with 3% Brewer's thiogly-collate broth. Three days later, peritoneal exudate cells were harvested and incubated. Two hours later, nonadherent cells were removed and the adherent monolayer cells were used as PMs. Human embryonic kidney (HEK293T) cells were obtained from American Type Culture Collection (Manassas, VA). The cells were cultured at 37 °C under 5% $CO_2$ in Dulbecco's Modified Eagle Medium supplemented with 10% fetal calf serum (Invitrogen-Gibco). The concentration of agonists or stimuli were used as below: LPS 200 ng/ml for mouse primary PMs, LPS 1 µg/ml for mouse embryonic fibroblasts (MEFs), ATP 2.5 mM, Nig 50 mM, MG132 10 µM, chloroquine 50 µM, 3-MA 10 mM. Poly(dA:dT) were transfected into macrophages with the final concentration as 200 ng/ml.

**Plasmid construction and transfection**. TRIM28 expression plasmid (SC321588) was purchased from Origene and then Flag-tag was added. TRIM28 C651A was generated using the KOD-Plus-Mutagenesis Kit (Toyobo, Osaka, Japan). Expression plasmids for NLRP3, ASC, Caspase-1, HA-Ub WT, mutant K48, and mutant K63 were described previously[10,12]. HA-SUMO1 (HG13095-NY), HA-SUMO2 (HG13443-NY), HA-SUMO3 (HG12782-NY), and His-UBC9 (HG143205-NH) were from Sino Biological. All constructs were confirmed by DNA sequencing. Plasmids were transiently transfected into HEK293T cell with Lipofectamine 2000 reagent (Invitrogen) according to the manufacturer's instructions.

**RNA interference assay**. siRNAs were synthesized as following: murine *Trim28*: 5′-GACAUCGUGGAGAAUUAUU-3′, negative control: 5′-UUCUCCGAACGUG UCACGU-3′. These siRNA duplexes were transfected into mouse PMs using INTERFERin reagents (Polyplus-transfection) according to the manufacturer's instructions.

**Enzyme-linked immunosorbent assay (ELISA)**. The concentrations of mouse IL-1β (cat. number: 1210123), TNF (cat. number: 1217202), and IL-6 (cat. number: 1210603) were measured using ELISA kits (Dakewe Biotech Company Ltd., Shenzhen, China) according to the manufacturer's instructions.

**RNA quantitation**. RNA from macrophages was extracted by using the RNA-fast200 RNA Extraction Kit according to the manufacturer's instructions. Quantitative real-time PCR analysis was carried out using the Applied Biosystems StepOnePlus Real-Time PCR System and SYBR RT-PCR kits (Roche). Specific primers used for real-time PCR assays are listed in Supplementary Table 1. Data are normalized to *β-Actin* expression in each sample.

**Immunofluorescent and confocal microscopy**. MEFs, which were plated on glass coverslips in 24-well plates, were fixed with Immunol Staining Fix Solution (Beyotime). Cells were permeabilized with 0.5% Triton-X 100 in phosphate-buffered saline (PBS) and blocked with Immunol Staining Blocking Buffer (Beyotime) for 1 h. Then the cells were incubated with anti-TRIM28 (Mouse) and anti-NLRP3 (Rabbit) diluted in Immunol Staining Primary Antibody Dilution Buffer (Beyotime) at 4 °C overnight. Next, the cells were stained with secondary Abs (anti-Mouse conjugated to Alexa Fluor 488 and anti-Rabbit conjugated to Alexa Fluor 633) and nuclei were stained with DAPI (Beyotime)[35]. Then the cells were subjected to microscopic analysis with the Zeiss LSM780 confocal laser microscope, which was provided by the Micro Characterization Facility of Shandong University.

**IP and immunoblot analysis**. For IP, whole-cell lysates were lysed in IP buffer containing 1.0% (vol/vol) Nonidet P 40, 50 mM Tris–HCl, pH 7.4, 50 mM EDTA, 150 mM NaCl, and a protease inhibitor "cocktail" (Sigma). After 15 min centrifugation at 12,000 × g at 4 °C, protein concentrations in the lysates were measured with a bicinchoninic acid assay (Pierce, Rockford, IL), and then supernatants were collected and incubated with protein G Plus-Agarose IP reagent together with specific antibody overnight. Beads were washed five times with IP buffer. Immunoprecipitates were eluted by boiling with 1% (wt/vol) SDS sample buffer. For immunoblot analysis, cells were lysed with RIPA Reagent (Pierce, Rockford, IL) supplemented with a protease inhibitor "cocktail." After 15 min centrifugation at 12,000 × g at 4 °C, protein concentrations in the lysates were measured with a bicinchoninic acid assay. Equal amounts of lysates were separated by SDS–PAGE and then were transferred onto nitrocellulose membranes for immunoblot analysis. Cell culture supernatants were harvested and concentrated for immunoblot with Amicon Ultra 10K from Millipore. NLRP3 expression in Chase assay was quantitated by measuring band intensities using the "ImageJ" software. The values were normalized to Actin.

**SUMOylation assays**. Cells cultured in 6-cm plates were transfected with the indicated plasmids (HEK293T cells) or primed with LPS and ATP (mouse PMs). Whole-cell lysates were lysed in 40 µl SDS lysis buffer containing 5.0% (vol/vol) Glycerol, 50 mM Tris–HCl, pH 6.8, 40 mM DTT, 1.0% (wt/vol) SDS, 20 µM NEM, and a protease inhibitor "cocktail." After 15 min centrifugation at 12,000 × g at 4 °C, lysates were put at 100 °C for 10 min. Protein concentrations in the lysates

were measured with a bicinchoninic acid assay and then diluted 1:10 with IP buffer and supernatants were collected and incubated with protein G Plus-Agarose IP reagent together with specific antibody overnight. Beads were washed five times with IP buffer. Immunoprecipitates were eluted by boiling with 1% (wt/vol) SDS sample buffer.

**In vivo LPS challenge**. WT or *Trim28*CKO mice (females, 6–8 weeks old) were i.p. injected with 25 mg/kg LPS or PBS. After 4–6 h, mice were sacrificed, and blood was collected for measurement of serum cytokines IL-1β, TNF, and IL-6 by ELISA.

**Statistical analysis**. Statistical significance between groups was determined by two-tailed Student's *t* test by using GraphPad Prism 6.0. Values of $p < 0.05$ were considered to be statistically significant.

**ImageJ software**. The protocol of ImageJ for western blotting: (1) Open an image. (2) Select the strips by rectangle. (3) Click Analyze-Gels-Select First Lane. (4) Click Analyze-Gels-Plot Lanes. (5) Seal off every area by straight line. (6) Select every area wand tool. (7) Get the results from a new window.

## Data availability

The authors declare that the data supporting the findings of this study are available within the paper and its supplementary information files or from the corresponding author upon reasonable request. Source data are provided with this paper.

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

## Acknowledgements

This work was supported by grants from the National Natural Science Foundation of China (81622030, 31570867, 31870866, and 81861130369). W.Z. is a Newton Advanced Fellow awarded by the Academy of Medical Sciences (NAF007\1003).

## Author contributions

W.Z. designed and supervised the research; Y.Q., Q.L., W.L., R.Y., L.T., M.J., and C.Z. performed the experiments; Y.Q. and W.Z analyzed the data and wrote the paper.

## Competing interests

The authors declare no competing interests.
