## [Peer Review File · Nature Communications]

REVIEWER COMMENTS

Reviewer #1 (Remarks to the Author):

The manuscript by Quin et al., describes the role of TRIM28, an E3 SUMO ligase, in catalyzing the binding of SUMO to NLRP3, thus promoting its stability due to inhibition of ubiquitin-proteasome proteolytic pathway. Different types of proofs are shown to demonstrate the conclusions and the manuscript is generally well-designed with reasonable conclusions. However many technical problems are present throughout the paper and some additional proof must be added to be fully convincing.

Comments

- 1) Please check the manuscript for typos and english mistakes (for example: "enhances" should be replaces with "enhancing" in line 56)
- 2) In the Introduction authors states that binding of NLRP3 with SUMO2/3 catalyzed by MUL1 "...restrains NLRP3 activation..." (line 65-66), whereas this sentence is contradicted by the sentence of lines 72-75 in which authors wrote that "...SUMO2/3 catalyzed SUMOylation of NLRP3....stabilizes NLRP3 protein expression and enhances inflammasome activation...". Would you please clarify this discrepancy on the role of SUMO2/3 in NLRP3 activation or inhibition?
- 3) Results of Figure 1:
 - a) the increased expression of TRIM28 upon stimulation of cells with LPS (fig. 1b) is the consequence of increased transcription? Please, answer to this by performing real-time experiments on TRIM28 expression.
 - b) The sequence of experiments in figure 1 should be changed. After the expression of TRIM28 (fig. 1b), the co-localization of TRIM28 with NLRP3 (current fig. 1e) should follows before the immunoprecipitation experiments.
 - c) IP of figure 1c lack the isotypic antibody negative control
 - d) How many experiments were performed? Are experiments shown in figure 1 representative of how many experiments?
- 4) Results of figure 2:
 - a) Please explain the meaning of AIM2 inflammasome activation (line 98)
 - b) Please explain the meaning of GSDMD (line 101) and N-GSGMD in the figure 2c
 - c) Correct the sentence "...To further the function of TRIM28..." (line 104)
 - d) It seems (deducing by number of circles) that the number of experiments in figure 2a, b, and d is 3. To perform an appropriate statistical analysis the number of experiments must at least 5.
 - e) In figure 2c, immunoblot analysis must be from different filters due to differences in bands' MW. Authors should clarify if lanes come from different immunoblot and, if this is the case, they should show an actine lane for each immunoblot (only one actine lane is shown in figure 2c).
 - f) Why the absence of TRIM28 does not completely eliminate NLRP3 expression? Is there any additional mechanism of NLRP3 expression beside TRIM28?
- 5) Results of figure 3:
 - a) Figure 3 should precede fig. 2, since priming come before activation
 - b) In figure 3a and c, immunoblot analyses must show also the total I κ B α , beside the pI κ B α . In addition, quantitation of bands from at least 5 experiments should be shown. Figure 3c should be moved to figure 3b (and viceversa the current 3b should be renamed as 3c)
 - c) It seems that 3 experiments are plotted in figure 3b and c, whereas I think that at least 5 experiments should be performed.
 - d) Results of figure 4: how many experiments were performed in a, b, c, d, f and g? the quantitation by measuring band intensities using "ImageJ" software should apply to all immunoblot analyses on at least 5 experiments. In addition immunoblots in figure 4b and c show a wide differences in MW (ranging from 15 KDA to 100 KDA) and this is not possible in only one filter. So please show the actin for each immunoblots performed. The software used for measuring band intensities should be described in "materials and Methods". The sentence "...Although TRIM28 has transcriptional regulatory functions¹⁸, it did not enhance NLRP3 transcription..." has been neither experimentally demonstrated nor demonstrated by a literature citation.
- 7) Results of figure 5: In the IP of figures 5a, b and c, the IP isotypic control was not shown and the input for ubiquitin was not shown either. Please, show also if NLRP3 migrate at the same level of ubiquitin bands in western blotting experiments. In suplementary figure 3, the IP isotypic control is not shown. Could you please explain the what K48- and K63-linked ubiquitins are?
- 8) Results of figure 6: In all IPs of figure 6, IP isotypic controls are absent. In addition, in fig 6a, Sumo1 input is lacked, in fig. 6b, Sumo2-3 input is not present and in figures 6d, e and f, His input is absent. In order to strenghten the notion that SUMO binds to NLRP3 and that UBC
- 9) 9 binds to NLRP3, please perform in vitro pull-down experiments to demonstrate their in vitro binding.

10) Discussion: the sentence in lines 198-200 should be english checked.

Authors stated that "...SUMO1 modification of NLRP3 may block the conjugation of ubiquitin competing the same K sites in NLRP3 and ...". Is the SUMO1/Ubiquitin conjugation site referred to K48? Please show in a figure of the paper the aminoacidic sequence of NLRP3 and highlight not only the lysine involved in the conjugation with SUMO1/Ubiquitin, but also the SUMO-consensus site (VKLD?)

11) Materials and Methods: Why mice used are both males and females? Do this introduce any possible variability in the experiments? You should indicate experiments performed with male or with female mice. Is t-student test the correct test for statistical significance? I think the minimum number of experiments should be increased up to five and the appropriate statistical test is the Mann-Withney or the Anova test.

Reviewer #2 (Remarks to the Author):

This paper identifies an E3 SUMO Ligase, TRIM28, as a SUMO ligase responsible for NLRP3 SUMOYLATION triggered by LPS and that contributes to the control of NLRP3 levels, hence contributing to NLRP3 inflammasome regulation This works adds significant knowledge to NLRP3 field and adds a new player in the regulation of NLRP3 by PTM at the priming step.

The authors validate their data in HEK293 systems but also in primary cells from TRIM28 Ko mice as well as TRIM28 silenced macrophages. They also show TRIM28 involvement in an in vivo experiment using TRIM28 deficient mice.

I have the following questions/comments to for the authors:

- Figure 1D. it looks like NLRP3 and TRIM28 interaction occurs equally in untreated and 4 and 8 h after LPS, despite levels of NLRP3 and TRIM28 increasing after LPS priming. What is your explanation for this? Does it mean that the levels of TRIM28 and NLRP3 are irrelevant for their interaction?
- Figure 1E. I am surprised that NLRP3 is so nuclear after LPS. Has this been described before or you think it is an artefact from the anti-NLRP3?
- Figure 2C. I know that you have checked IL-6 and TNF, but have you checked that Pro-IL1B levels are not changed by TRIM28? Maybe just run the WB for anti-il1b too.
- Figure 4e. Is that the densitometry from blots? Please indicate what the Y axis is.
- Figure 5B. Where it says vector, do you mean empty vector (no Ub) or WT ubiquitin? Specify in the fig.
- Figure 5c and d. The expression of the inactive TRIM28 seems to be lower than TRIM28 WT. Although I believe this might not be the reason to detect more ubiquitination, I think you need to show a experiment where the levels of TRIM28 (WT and mutant) are similar to then evaluate effect on ubiquitination. Do you think the catalytically dead TRIM28 is less stable?
- Figure 6B. I find hard to see that there is a decrease in sumoylation (SUMO2/3) after ATP.
- Also, have you checked if there is an increase/decreased NLRP3-TRIM28 interaction after ATP? Do you know/think if TRIM28 also plays a direct role on the activation step?
- In the supplementary fig 2 it looks like in the TRIM28-CKO you still detect a band for TRIM28 (lanes 4, 7 and 8). Is this really a knock out or a knockdown? And how can you explain this?
- In general in any NLRP immunoprecipitation you show NLRP3 in the input (this is missing in several blots).

Reviewer #1:

The manuscript by Qin et al., describes the role of TRIM28, an E3 SUMO ligase, in catalyzing the binding of SUMO to NLRP3, thus promoting its stability due to inhibition of ubiquitin-proteasome proteolytic pathway. Different types of proofs are shown to demonstrate the conclusions and the manuscript is generally well-designed with reasonable conclusions. However many technical problems are present throughout the paper and some additional proof must be added to be fully convincing.

Answer: We appreciate very much for your work in reviewing our manuscript. In accordance with your insightful and valuable suggestions and comments, we carefully revised the manuscript and several additional experiments were performed. The point-by-point answers to the comments and suggestions were listed as below.

Comments

1) Please check the manuscript for typos and English mistakes (for example: “enhances” should be replaces with “enhancing” in line 56).

Answer: We are very sorry for the language mistakes. We have checked the article carefully and corrected the mistakes in the revised manuscript.

2) In the Introduction authors states that binding of NLRP3 with SUMO2/3 catalyzed by MUL1 “...restrains NLRP3 activation...” (line 65-66), whereas this sentence is contradicted by the sentence of lines 72-75 in which authors wrote that “...SUMO2/3 catalyzed SUMOylation of NLRP3....stabilizes NLRP3 protein expression and enhances inflammasome activation...”. Would you please clarify this discrepancy on the role of SUMO2/3 in NLRP3 activation or inhibition?

Answer: This is an important question. Although MUL1 mediated SUMO2/3 catalyzed SUMOylation of NLRP3 inhibited inflammasome activation, we found that TRIM28 mediated SUMO2/3 catalyzed SUMOylation facilitated inflammasome activation. In addition, TRIM28, but not MUL1, inhibited the ubiquitination degradation of NLRP3. Therefore, TRIM28 and MUL1 regulate NLRP3 activity by different mechanisms. Multiple potential SUMOylation sites exist in NLRP3 protein (such as K204 and K689, etc.), and TRIM28 and MUL1 may mediate NLRP3 SUMOylation at different lysine sites. SUMOylation mediated by TRIM28 may block the ubiquitination site of NLRP3, and thus inhibits NLRP3 proteasomal degradation. Further analysis of the SUMOylation and ubiquitination sites of

NLRP3 will help to elucidate the underlying mechanisms. We discussed this issue in Discussion in the revised manuscript.

3) Results of Figure 1:

a) the increased expression of TRIM28 upon stimulation of cells with LPS (fig. 1b) is the consequence of increased transcription? Please, answer to this by performing real-time experiments on TRIM28 expression.

Answer: We appreciated the valuable suggestions and examined *Trim28* mRNA level. We found that the mRNA levels of *Trim28* decreased first and then increased with LPS stimulation (Response Figure.1). We speculate that there are other regulatory mechanisms for TRIM28 protein levels in response to LPS stimulation. As the protein level of TRIM28 is crucial for its function, we did not include these data in the revised manuscript.

Response Figure.1 RT-PCR analysis of *Trim28* mRNA in LPS-stimulated mouse PMs.

The sequences of primers used for RT-PCR were 5'- CACACTCACCTGCCGCGA-3', 5'-TCAACCTGCACTCGCTTCTG-3'.

b) The sequence of experiments in figure 1 should be changed. After the expression of TRIM28 (fig. 1b), the co-localization of TRIM28 with NLRP3 (current fig. 1e) should follow before the immunoprecipitation experiments.

Answer: We appreciated the valuable suggestions and revised accordingly in figure 1.

c) IP of figure 1c lack the isotypic antibody negative control

Answer: We appreciated the suggestion. In the present study, the transfection of only one plasmid used as a negative control.

d) How many experiments were performed? Are experiments shown in figure 1 representative of how many experiments?

Answer: Similar results were obtained from three independent experiments. The experimental results presented are representative. We included these information in the Figure Legends in the revised manuscript.

4) Results of figure 2:

a) Please explain the meaning of AIM2 inflammasome activation (line 98)

b) Please explain the meaning of GSDMD (line 101) and N-GSGMD in the figure 2c

Answer: We appreciated for the valuable suggestion and provided the information in the revised manuscript.

c) Correct the sentence "...To further the function of TRIM28..." (line 104)

Answer: We appreciated for the valuable suggestion and revised this issue in the manuscript.

d) It seems (deducing by number of circles) that the number of experiments in figure 2a, b, and d is 3. To perform an appropriate statistical analysis the number of experiments must at least 5.

Answer: Thanks for the suggestions. Three samples are sufficient for the statistical analysis.

e) In figure 2c, immunoblot analysis must be from different filters due to differences in bands' MW. Authors should clarify if lanes come from different immunoblot and, if this is the case, they should show an actin lane for each immunoblot (only one actin lane is shown in figure 2c).

Answer: We appreciated for the valuable suggestion. These lanes came from different immunoblot, but came from the same set of samples. The actin bands for each immunoblot were shown in the revised manuscript (Fig. 3c).

Fig. 3c

f) Why the absence of TRIM28 does not completely eliminate NLRP3 expression? Is there any additional mechanism of NLRP3 expression beside TRIM28?

Answer: TRIM28 inhibits proteasomal degradation of NLRP3, and then promotes NLRP3 expression. However, in *Trim28* deficiency macrophages, NLRP3 is still synthesized normally, and NLRP3 degradation is enhanced. Therefore, the absence of TRIM28 does not completely eliminate NLRP3 expression.

5) Results of figure 3:

a) Figure 3 should precede fig. 2, since priming come before activation

Answer: We accepted the valuable suggestions and revised accordingly.

b) In figure 3a and c, immunoblot analyses must show also the total IκBα, beside the pIκBα. In addition, quantitation of bands from at least 5 experiments should be shown.

Answer: We appreciated the valuable suggestions. Total IκBα and p-IκBα from 5 experiments were shown below. Both *Trim28* deficiency and knockdown had no effects on the total IκBα and LPS-induced phosphorylation of IκB-α. However, the dynamic changes of total IκBα and p-IκBα were different. So, we did not pool these data together and calculate the mean quantitation of bands. We presented the representative results and added these new data in the revised manuscript.

Figure 3c should be moved to figure 3b (and vice versa the current 3b should be renamed as 3c)

Answer: We accepted the suggestions and revised this issue in the manuscript.

c) It seems that 3 experiments are plotted in figure 3b and c, whereas I think that at least 5 experiments should be performed.

Answer: We appreciated very much for the comments. We believe that three samples are sufficient for the statistical analysis.

6) Results of figure 4: how many experiments were performed in a, b, c, d, f and g? the quantitation by measuring band intensities using "ImageJ" software should apply to all immunoblot analyses on at least 5 experiments. In addition immunoblots in figure 4b and c show a wide differences in MW (ranging from 15 KDA to 100 KDA) and this is not possible in only one filter. So please show the actin for each immunoblots performed.

Answer: We are sorry for lack of the key information and we added them in the revised manuscript. Similar results were obtained from three independent experiments in Fig.4. In Fig.4e, the quantitation of NLRP3 expression was measuring by “ImageJ” software and the experiments were repeated 5 times. The bands of Fig.4b did not overlap. To better present Fig.4c, we showed additional actin. These bands of Fig.4c came from different immunoblots, and all bands except Caspase-1 came from the same group of samples. To prevent any misunderstanding, we replaced with Caspase-1 from the same group of samples. The actin bands for each immunoblot are shown in the revised manuscript.

The software used for measuring band intensities should be described in “materials and Methods”.

Answer: We added the description of “ImageJ” software in “Materials and Methods”.

The sentence “...Although TRIM28 has transcriptional regulatory functions¹⁸, it did not enhance NLRP3 transcription...” has been neither experimentally demonstrated nor demonstrated by a literature citation.

Answer: The sentence “...Although TRIM28 has transcriptional regulatory functions¹⁸, it did not enhance NLRP3 transcription...” has been experimentally demonstrated in fig.2b, 2d. We added the information in the revised manuscript.

7) Results of figure 5: In the IP of figures 5a, b and c, the IP isotypic control was not shown and the input for ubiquitin was not shown either. Please, show also if NLRP3 migrate at the same level of

ubiquitin bands in western blotting experiments. In supplementary figure 3, the IP isotypic control is not shown.

Answer: We appreciated for the valuable suggestion. We repeated the experiments of Fig.5 and included the suggested controls. In addition, we added the lacked input Transfection of only one plasmid can be used as a negative control. Therefore, in figure 5b and supplemental figure 3, the transfection of only one plasmid was used as negative control. We changed the presented figures in Fig.5 in the revised manuscript.

Fig. 5

Could you please explain what K48- and K63-linked ubiquitins are?

Answer: Ubiquitin mutant vectors K48-Ub and K63-Ub, contain arginine substitutions of all of its lysine residues except the one at position 48 and 63 respectively. We provided the description of K48- and K63-linked ubiquitins in the revised manuscript.

8) Results of figure 6: In all IPs of figure 6, IP isotypic controls are absent. In addition, in fig 6a, Sumo1 input is lacked, in fig. 6b, Sumo2-3 input is not present and in figures 6d, e and f, His input is absent.

Answer: We appreciated for the valuable suggestion. We repeated the experiments of Fig.6 and included the suggested controls. In addition, we added the lacked input. We changed the presented figures in Fig.6 in the revised manuscript.

Fig. 6

In order to strengthen the notion that SUMO binds to NLRP3 and that UBC9 binds to NLRP3, please perform in vitro pull-down experiments to demonstrate their in vitro binding.

Answer: This is a great question. The in vitro binding between NLRP3 and SUMO, UBC9 and NLRP3 was confirmed previously (Shao Luyao, et al. *FASEB J*, 2020, 34: 1497-1515.) and we cited the reference in the revised manuscript.

[REDACTED]

(Shao Luyao, et al. *FASEB J*, 2020, 34: 1497-1515.)

9) *Discussion: the sentence in lines 198-200 should be English checked. Authors stated that "...SUMO1 modification of NLRP3 may block the conjugation of ubiquitin competing the same K sites in NLRP3 and ...". Is the SUMO1/Ubiquitin conjugation site referred to K48? Please show in a figure of the paper the aminoacidic sequence of NLRP3 and highlight not only the lysine involved in the conjugation with SUMO1/Ubiquitin, but also the SUMO-consensus site (VKLD?)*

Answer: We are sorry for the unclear statement. The lysine (K) sites refer to the amino acids of NLRP3 protein (but not K48), which could be ubiquitinated. To better indicate the potential mechanisms, we revised the work model (Fig.7) and added the information.

10) *Materials and Methods: Why mice used are both males and females? Do this introduce any possible variability in the experiments? You should indicate experiments performed with male or with female mice.*

Answer: We performed all the experiments with mice of different genders (we include the information in the Mice section of the Methods), and found no differences between male and female. We indicate the gender in the *in vivo* experiments, in which we used female mice. We provide these information in

the revised manuscript.

Is t-student test the correct test for statistical significance? I think the minimum number of experiments should be increased up to five and the appropriate statistical test is the Mann-Withney or the Anova test.

Answer: Thanks for the comments. The ANOVA test is suitable for multiple samples analysis. The Mann-Withney Text is suitable for non-parametric tests. But our experiments are all comparisons between two groups of samples. Similar examples analyses were also used the t-student test instead of the Mann-Withney test (such as Ising Christina, et al. *Nature*, 2019, 575: 669-673.; Chen Jueqi, et al. *Nature*, 2018, 564: 71-76.; Jia Mutian, et al. *Nat Immunol*, 2020, 21: 727-735.). Then, the t-student test is suitable for the statistical analysis in the experiments. In addition, three samples are sufficient for the statistical analysis with t-test.

Reviewer #2:

This paper identifies an E3 SUMO Ligase, TRIM28, as a SUMO ligase responsible for NLRP3 SUMOylation triggered by LPS and that contributes to the control of NLRP3 levels, hence contributing to NLRP3 inflammasome regulation. This work adds significant knowledge to NLRP3 field and adds a new player in the regulation of NLRP3 by PTM at the priming step. The authors validate their data in HEK293 systems but also in primary cells from TRIM28 Ko mice as well as TRIM28 silenced macrophages. They also show TRIM28 involvement in an in vivo experiment using TRIM28 deficient mice. I have the following questions/comments to for the authors:

Answer: We appreciate very much for your time in reviewing our manuscript. In accordance with your insightful and valuable suggestions and comments, we carefully revised the manuscript and several additional experiments were performed. The point-by-point answers to the comments and suggestions were listed as below.

- Figure 1D. it looks like NLRP3 and TRIM28 interaction occurs equally in untreated and 4 and 8 h after LPS, despite levels of NLRP3 and TRIM28 increasing after LPS priming. What is your explanation for this? Does it mean that the levels of TRIM28 and NLRP3 are irrelevant for their interaction?

Answer: Yes, this is an important question. In Fig.1d, the cell lysates were not quantified. To better clarify the question, we repeated the IP experiment after quantification of cell lysates. As shown below, the interaction between NLRP3 and TRIM28 increased after LPS stimulation.

Response Figure.2 IP analysis of endogenous association between TRIM28 and NLRP3 in LPS-stimulated mouse PMs.

- Figure 1E. I am surprised that NLRP3 is so nuclear after LPS. Has this been described before or you

think it is an artefact from the anti-NLRP3?

Answer: We performed this experiment multiple times. Each time, NLRP3 localized in both cytoplasm and nucleus in LPS-stimulated MEFs. It has been described that NLRP3 is localized in the nucleus in LPS-stimulated MEFs (Song, H. et al. *Nat Commun*, 2016: 7, 13727.) and BMEM (V. G. Magupalli et al. *Science*, 2020, 369: eaas8995; Samir P., et al. *Nature*, 2019, 573: 590-594.).

(Song, H. et al. *Nat Commun*, 2016: 7, 13727.)

Editorial Note: the authors wish to note that the above caption should instead cite Figure 5 a&b from Song, H. et al., *Nat. Commun.* 1, 6042 (2020) <https://doi.org/10.1038/s41467-020-19939-8>

[REDACTED]

(V. G. Magupalli et al., *Science*, 2020, 369: eaas8995)

[REDACTED]

(Samir P., et al. *Nature*, 2019, 573: 590-594.).

- Figure 2C. I know that you have checked IL-6 and TNF, but have you checked that Pro-IL1B levels are not changed by TRIM28? Maybe just run the WB for anti-il1b too.

Answer: We accepted the valuable suggestions performed additional experiments accordingly. *Trim28* deficiency had no effects on pro-IL-1 β level (Fig.3C). We added these new data in the revised manuscript.

Fig. 3c

- Figure 4e. Is that the densitometry from blots? Please indicate what the Y axis is.

Answer: Yes, Fig.4e indicates the densitometry from blots. The protein expression levels of NLRP3 and Actin in fig.4d were quantitated by measuring band intensities using 'ImageJ' software, and the values of NLRP3 were normalized to Actin. The Y-axis values are the ratio of all the values to the value without the stimulus. The Y axis is relative expression of NLRP3. We added the information in Fig.4e.

- Figure 5B. Where it says vector, do you mean empty vector (no Ub) or WT ubiquitin? Specify in the fig. **Answer:** We are sorry for the unclear description. It is WT ubiquitin in Fig.5b. We specified it in Fig.5b in the revised manuscript.

- Figure 5c and d. The expression of the inactive TRIM28 seems to be lower than TRIM28 WT. Although I believe this might not be the reason to detect more ubiquitination, I think you need to show an experiment where the levels of TRIM28 (WT and mutant) are similar to then evaluate effect on ubiquitination. Do you think the catalytically dead TRIM28 is less stable?

Answer: We accepted the valuable suggestions. We re-extracted the plasmids and then repeated the experiments. In the new Fig.5c and d, the expression of inactive TRIM28 is similar to WT TRIM28. Inactive TRIM28 (C651) lost the ability to inhibit NLRP3 ubiquitination and degradation (Fig. 5c and 5d).

Fig. 5

- Figure 6B. I find hard to see that there is a decrease in sumoylation (SUMO2/3) after ATP.

Answer: We repeated the experiments and changed Fig.6b with new data in the revised manuscript.

Fig. 6b

- Also, have you checked if there is an increase/decreased NLRP3-TRIM28 interaction after ATP? Do you know/think if TRIM28 also plays a direct role on the activation step?

Answer: This is a great question and we performed additional experiments accordingly to the suggestions. An association between TRIM28 and NLRP3 was detected in both resting LPS-primed and ATP-activated mouse PMs (Fig. 1d). However, the interaction between TRIM28 and NLRP3 did not change after ATP stimulation. We added these new data in the revised manuscript. *Trim28* deficiency attenuates NLRP3 inflammasome activation *in vivo* and *in vitro* (Fig. 3a and e). *Trim28* deficiency attenuated SUMO1 modification of NLRP3 in macrophages after NLRP3 inflammasome activation (Fig. 6a). Therefore, we concluded that TRIM28 facilitated NLRP3 inflammasome activation, by enhancing NLRP3 protein expression, in both priming and activation phases.

Fig. 1d

- In the supplementary fig 2 it looks like in the TRIM28-CKO you still detect a band for TRIM28 (lanes 4, 7 and 8). Is this really a knock out or a knockdown? And how can you explain this?

Answer: Mice deficient in *Trim28* die prior to gastrulation. We thus crossed *Trim28*^{fl^{ox}/fl^{ox}} mice with *Lyz2*^{cre} transgenic mice to specifically knockout *Trim28* in myeloid cells. Sometimes, the efficiency of conditional knockout is less than 100%. We considered that the knockout efficiency was sufficient for this study.

- In general in any NLRP immunoprecipitation you show NLRP3 in the input (this is missing in several blots).

Answer: We appreciated the valuable suggestions. We repeated the related experiments and examined NLRP3 expression in the input. We added these new data in the revised manuscript.

Fig. 5a

Fig. 6a

REVIEWERS' COMMENTS

Reviewer #1 (Remarks to the Author):

Authors have improved very much the manuscript by performing most of the suggested changes. However, authors have ignored some suggestions and manuscript still lack some suggested necessary modifications.

3. Results of Figure 1:

a) the increased expression of TRIM28 upon stimulation of cells with LPS (fig. 1b) is the consequence of increased transcription? Please, answer to this by performing real-time experiments on TRIM28 expression.

Answer: We appreciated the valuable suggestions and examined Trim28 mRNA level. We found that the mRNA levels of Trim28 decreased first and then increased with LPS stimulation (Response Figure.1). We speculate that there are other regulatory mechanisms for TRIM28 protein levels in response to LPS stimulation. As the protein level of TRIM28 is crucial for its function, we did not include these data in the revised manuscript.

Additional comment.

Real-time experiment results are very informative and I think authors should include these data in the revised manuscript (at least in supplementary data);

c) IP of figure 1c lack the isotypic antibody negative control

Answer: We appreciated the suggestion. In the present study, the transfection of only one plasmid used as a negative control.

Additional comment.

Is the isotype of anti-flag the same as the isotype of anti-myc? If so, isotype should be specified in "materials and methods";

d) How many experiments were performed? Are experiments shown in figure 1 representative of how many experiments?

Answer: Similar results were obtained from three independent experiments. The experimental results presented are representative. We included these information in the Figure Legends in the revised manuscript.

Additional comment.

Three experiments are insufficient for an appropriate statistical analysis: authors should perform at least 5 experiments.

The meaning of IP (Immunoprecipitation) should be show up at its first appearance (in (d) rather than in (e)).

4) Results of figure 2:

a) Please explain the meaning of AIM2 inflammasome activation (line 98)

b) Please explain the meaning of GSDMD (line 101) and N-GSDMD in the figure 2c

Answer: We appreciated for the valuable suggestion and provided the information in the revised manuscript.

Additional comment.

I still don't understand what is AIM2 inflammasome: could authors clarify it? Could you report the complete meaning of GSDM (GSDM like AIM2 are abbreviations)?

d) It seems (deducing by number of circles) that the number of experiments in figure 2a, b, and d is 3. To perform an appropriate statistical analysis the number of experiments must at least 5.

Answer: Thanks for the suggestions. Three samples are sufficient for the statistical analysis.

Additional comment.

I think five rather than three experiments are sufficient for the statistical analysis.

f) Why the absence of TRIM28 does not completely eliminate NLRP3 expression? Is there any additional mechanism of NLRP3 expression beside TRIM28?

Answer: TRIM28 inhibits proteasomal degradation of NLRP3, and then promotes NLRP3 expression. However, in Trim28 deficiency macrophages, NLRP3 is still synthesized normally, and NLRP3 degradation is enhanced. Therefore, the absence of TRIM28 does not completely eliminate NLRP3 expression.

Additional comment.

I think authors should report this explanation in the text.

5) Results of figure 3:

b) In figure 3a and c, immunoblot analyses must show also the total IκBα, beside the pIκBα. In

addition, quantitation of bands from at least 5 experiments should be shown.

Answer: We appreciated the valuable suggestions. Total I κ B α and p-I κ B α from 5 experiments were shown below. Both Trim28 deficiency and knockdown had no effects on the total I κ B α and LPS-induced phosphorylation of I κ B- α . However, the dynamic changes of total I κ B α and p-I κ B α were different. So, we did not pool these data together and calculate the mean quantitation of bands. We presented the representative results and added these new data in the revised manuscript.

Additional comment.
Despite authors affirm that "Total I κ B α and p-I κ B α from 5 experiments were shown below", in the figure's legend is still reported "data from three experiments".

c) It seems that 3 experiments are plotted in figure 3b and c, whereas I think that at least 5 experiments should be performed.

Answer: We appreciated very much for the comments. We believe that three samples are sufficient for the statistical analysis.

Additional comment.

I believe five rather than three samples are sufficient for the statistical analysis.

6) Results of figure 4: how many experiments were performed in a, b, c, d, f and g? the quantitation by measuring band intensities using "ImageJ" software should apply to all immunoblot analyses on at least 5 experiments. In addition immunoblots in figure 4b and c show a wide differences in MW (ranging from 15 KDA to 100 KDA) and this is not possible in only one filter. So please show the actin for each immunoblots performed.

Answer: We are sorry for lack of the key information and we added them in the revised manuscript. Similar results were obtained from three independent experiments in Fig.4. In Fig.4e, the quantitation of NLRP3 expression was measuring by "ImageJ" software and the experiments were repeated 5 times. The bands of Fig.4b did not overlap. To better present Fig.4c, we showed additional actin. These bands of Fig.4c came from different immunoblots, and all bands except Caspase-1 came from the same group of samples. To prevent any misunderstanding, we replaced with Caspase-1 from the same group of samples. The actin bands for each immunoblot are shown in the revised manuscript.

Additional comment.

"Similar results were obtained from three independent experiments in Fig.4": I think five independent experiments are needed.

I don't understand the meaning of "The bands of Fig.4b did not overlap".

7) Results of figure 5: In the IP of figures 5a, b and c, the IP isotypic control was not shown and the input for ubiquitin was not shown either. Please, show also if NLRP3 migrate at the same level of ubiquitin bands in western blotting experiments. In supplementary figure 3, the IP isotypic control is not shown.

Answer: We appreciated for the valuable suggestion. We repeated the experiments of Fig.5 and included the suggested controls. In addition, we added the lacked input Transfection of only one plasmid can be used as a negative control. Therefore, in figure 5b and supplemental figure 3, the transfection of only one plasmid was used as negative control. We changed the presented figures in Fig.5 in the revised manuscript.

Additional comment.

Clarify the isotype of anti-flag and anti-myc (authors can write it in "material & methods section).

9) Discussion: the sentence in lines 198-200 should be English checked. Authors stated that "...SUMO1 modification of NLRP3 may block the conjugation of ubiquitin competing the same K sites in NLRP3 and ...". Is the SUMO1/Ubiquitin conjugation site referred to K48? Please show in a figure of the paper the aminoacidic sequence of NLRP3 and highlight not only the lysine involved in the conjugation with SUMO1/Ubiquitin, but also the SUMO-consensus site (VKLD?)

Answer: We are sorry for the unclear statement. The lysine (K) sites refer to the amino acids of NLRP3 protein (but not K48), which could be ubiquitinated. To better indicate the potential mechanisms, we revised the work model (Fig.7) and added the information.

Additional comment.

Authors did not show the amino acidic sequence of NLRP3 and did not highlight the SUMO-consensus site.

The work model seems to me basically the same as in the first version of the manuscript.

Is t-student test the correct test for statistical significance? I think the minimum number of experiments should be increased up to five and the appropriate statistical test is the Mann-Whitney or the Anova test.

Answer: Thanks for the comments. The ANOVA test is suitable for multiple samples analysis. The

Mann-Whitney Test is suitable for non-parametric tests. But our experiments are all comparisons between two groups of samples. Similar examples analyses were also used the t-student test instead of the Mann-Whitney test (such as Ising Christina, et al. Nature, 2019, 575: 669-673.; Chen Jueqi, et al. Nature, 2018, 564: 71-76.; Jia Mutian, et al. Nat Immunol, 2020, 21: 727-735.). Then, the t-student test is suitable for the statistical analysis in the experiments. In addition, three samples are sufficient for the statistical analysis with t-test.
Additional comment.
Authors should perform five experiments.

Reviewer #2 (Remarks to the Author):

I would like to thank the authors for addressing the suggested comments. There are a couple of things that I would like to add.

1. I think the Immunofluorescence in figure 1 data does not add much. In the examples you show from previous publications sometimes NLRP3 is seen in the nucleus and sometimes not. As this figure does not really add much to your discoveries. I would particularly remove it and re-arrange the figure in the order suggested by the other reviewer.
2. In line 115 "...Trim28 knockdown also reduced IL-1 β secretion following LPS priming and ATP activation in mouse" By knockdown you mean siRNA? Maybe just clarify that.

Otherwise I am happy with the manuscript.

Reviewer #1 (Remarks to the Author):

Authors have improved very much the manuscript by performing most of the suggested changes. However, authors have ignored some suggestions and manuscript still lack some suggested necessary modifications.

Answer: We appreciate very much for your work in reviewing our manuscript. In accordance with your insightful and valuable suggestions and comments, we carefully revised the manuscript. The point-by-point answers to the comments and suggestions were listed as below.

3. Results of Figure 1:

a) the increased expression of TRIM28 upon stimulation of cells with LPS (fig. 1b) is the consequence of increased transcription? Please, answer to this by performing real-time experiments on TRIM28 expression.

Answer: We appreciated the valuable suggestions and examined Trim28 mRNA level. We found that the mRNA levels of Trim28 decreased first and then increased with LPS stimulation (Response Figure.1). We speculate that there are other regulatory mechanisms for TRIM28

protein levels in response to LPS stimulation. As the protein level of TRIM28 is crucial for its function, we did not include these data in the revised manuscript.

Additional comment.

Real-time experiment results are very informative and I think authors should include these data in the revised manuscript (at least in supplementary data);

Answer: We included the result as the supplementary data in the revised manuscript.

c) IP of figure 1c lack the isotypic antibody negative control

Answer: We appreciated the suggestion. In the present study, the transfection of only one plasmid used as a negative control.

Additional comment.

Is the isotype of anti-flag the same as the isotype of anti-myc? If so, isotype should be specified in “materials and methods”;

Answer: We accepted the valuable suggestions and provided the information in the “materials and methods”.

d) How many experiments were performed? Are experiments shown in figure 1 representative of how many experiments?

Answer: Similar results were obtained from three independent experiments. The experimental results presented are representative. We included these information in the Figure Legends in the revised manuscript.

Additional comment.

Three experiments are insufficient for an appropriate statistical analysis: authors should perform at least 5 experiments.

The meaning of IP (Immunoprecipitation) should be show up at its first appearance (in (d) rather than in (e)).

Answer: We appreciated very much for the comments. Similar results were obtained from three independent experiments. The experimental results presented are representative. We submitted original data from three independent experiments. The manuscript has been revised.

4) Results of figure 2:

a) Please explain the meaning of AIM2 inflammasome activation (line 98)

b) Please explain the meaning of GSDMD (line 101) and N-GSDMD in the figure 2c

Answer: We appreciated for the valuable suggestion and provided the information in the revised manuscript.

Additional comment.

I still don't understand what is AIM2 inflammasome: could authors clarify it? Could you report the complete meaning of GSDMD (GSDMD like AIM2 are abbreviations)?

Answer: AIM2 inflammasome is similar to NLRP3 inflammasome and consists of AIM2, ASC, and caspase1. The activation of AIM2 inflammasome also results in the cleaving of pro-IL-1 β for release. Unlike NLRP3 inflammasomes, it can only be activated by DNA (e.g., poly(dA:dT)) and cannot be activated by NLRP3 inflammasome activators.

The complete meaning of GSDMD is gasdermin D.

d) It seems (deducing by number of circles) that the number of experiments in figure 2a, b, and d is 3. To perform an appropriate statistical analysis the number of experiments must at least 5.

Answer: Thanks for the suggestions. Three samples are sufficient for the statistical analysis.

Additional comment.

I think five rather than three experiments are sufficient for the statistical analysis.

Answer: Thanks for the suggestions. Identical number of replicates had been reported in different research articles (e.g. Zhong Zhenyu, et al. *Nature*, 2018, 560: 198-203.; Chen Jueqi, et al. *Nature*, 2018, 564: 71-76.; Jia Mutian, et al. *Nat Immunol*, 2020, 21: 727-735). In our paper, the experiments were conducted three times. We consider this to be a sufficient sample size for demonstrating reproducibility of our findings.

f) Why the absence of TRIM28 does not completely eliminate NLRP3 expression? Is there any additional mechanism of NLRP3 expression beside TRIM28?

Answer: TRIM28 inhibits proteasomal degradation of NLRP3, and then promotes NLRP3

expression. However, in Trim28 deficiency macrophages, NLRP3 is still synthesized normally, and NLRP3 degradation is enhanced. Therefore, the absence of TRIM28 does not completely eliminate NLRP3 expression.

Additional comment.

I think authors should report this explanation in the text.

Answer: We appreciated very much for the comments. We added this explanation in the revised manuscript.

5) Results of figure 3:

b) In figure 3a and c, immunoblot analyses must show also the total I κ B α , beside the pI κ B α . In addition, quantitation of bands from at least 5 experiments should be shown.

Answer: We appreciated the valuable suggestions. Total I κ B α and p-I κ B α from 5 experiments were shown below. Both Trim28 deficiency and knockdown had no effects on the total I κ B α and LPS-induced phosphorylation of I κ B- α . However, the dynamic changes of total I κ B α and p-I κ B α were different. So, we did not pool these data together and calculate the mean quantitation of bands. We presented the representative results and added these new data in the revised manuscript.

Additional comment.

Despite authors affirm that “Total I κ B α and p-I κ B α from 5 experiments were shown below”, in the figure’s legend is still reported “data from three experiments”.

Answer: We appreciated very much for the comments. The manuscript was revised accordingly.

c) It seems that 3 experiments are plotted in figure 3b and c, whereas I think that at least 5 experiments should be performed.

Answer: We appreciated very much for the comments. We believe that three samples are sufficient for the statistical analysis.

Additional comment.

I believe five rather than three samples are sufficient for the statistical analysis.

Answer: We appreciated very much for the comments. Identical number of replicates had

been reported in different research articles (e.g. Zhong Zhenyu, et al. *Nature*, 2018, 560: 198-203.; Chen Jueqi, et al. *Nature*, 2018, 564: 71-76.; Jia Mutian, et al. *Nat Immunol*, 2020, 21: 727-735). In our paper, the experiments were conducted three times with highly reproducible. We consider this to be a sufficient sample size for demonstrating reproducibility of our findings. Thanks again for your precious suggestions.

6) Results of figure 4: how many experiments were performed in a, b, c, d, f and g? the quantitation by measuring band intensities using “ImageJ” software should apply to all immunoblot analyses on at least 5 experiments. In addition immunoblots in figure 4b and c show a wide differences in MW (ranging from 15 KDA to 100 KDA) and this is not possible in only one filter. So please show the actin for each immunoblots performed.

Answer: We are sorry for lack of the key information and we added them in the revised manuscript. Similar results were obtained from three independent experiments in Fig.4. In Fig.4e, the quantitation of NLRP3 expression was measuring by “ImageJ” software and the experiments were repeated 5 times. The bands of Fig.4b did not overlap. To better present Fig.4c, we showed additional actin. These bands of Fig.4c came from different immunoblots, and all bands except Caspase-1 came from the same group of samples. To prevent any misunderstanding, we replaced with Caspase-1 from the same group of samples. The actin bands for each immunoblot are shown in the revised manuscript.

Additional comment.

“Similar results were obtained from three independent experiments in Fig.4”: I think five independent experiments are needed.

I don't understand the meaning of “The bands of Fig.4b did not overlap”.

Answer: We appreciated very much for the comments. Similar results were obtained from three independent experiments. The experimental results presented are representative. We submitted original data from three experiments.

Fig.4b was from one filter.

7) Results of figure 5: In the IP of figures 5a, b and c, the IP isotypic control was not shown and the input for ubiquitin was not shown either. Please, show also if NLRP3 migrate at the

same level of ubiquitin bands in western blotting experiments. In supplementary figure 3, the IP isotopic control is not shown.

Answer: We appreciated for the valuable suggestion. We repeated the experiments of Fig.5 and included the suggested controls. In addition, we added the lacked input Transfection of only one plasmid can be used as a negative control. Therefore, in figure 5b and supplemental figure 3, the transfection of only one plasmid was used as negative control. We changed the presented figures in Fig.5 in the revised manuscript.

Additional comment.

Clarify the isotype of anti-flag and anti-myc (authors can write it in "material & methods section).

Answer: We indicated this in the "Materials and Methods".

9) Discussion: the sentence in lines 198-200 should be English checked. Authors stated that "...SUMO1 modification of NLRP3 may block the conjugation of ubiquitin competing the same K sites in NLRP3 and ...". Is the SUMO1/Ubiquin conjugation site referred to K48? Please show in a figure of the paper the aminoacidic sequence of NLRP3 and highlight not only the lysine involved in the conjugation with SUMO1/Ubiquitin, but also the SUMO-consensus site (VKLD?)

Answer: We are sorry for the unclear statement. The lysine (K) sites refer to the amino acids of NLRP3 protein (but not K48), which could be ubiquitinated. To better indicate the potential mechanisms, we revised the work model (Fig.7) and added the information.

Additional comment.

Authors did not show the amino acidic sequence of NLRP3 and did not highlight the SUMO-consensus site.

Answer: We appreciated very much for the comments. The model shows that SUMOylation and ubiquitination competitively bind to the same site. We did not specifically detect the specific amino acidic sites where NLRP3 is SUMOylated in this study, and we were unable to draw the amino acidic sequence of NLRP3.

The work model seems to me basically the same as in the first version of the manuscript.

Is t-student test the correct test for statistical significance? I think the minimum number of experiments should be increased up to five and the appropriate statistical test is the Mann-Withney or the Anova test.

Answer: Thanks for the comments. The ANOVA test is suitable for multiple samples analysis. The Mann-Withney Text is suitable for non-parametric tests. But our experiments are all comparisons between two groups of samples. Similar examples analyses were also used the t-student test instead of the Mann-Withney test (such as Ising Christina, et al. Nature, 2019, 575: 669-673.; Chen Jueqi, et al. Nature, 2018, 564: 71-76.; Jia Mutian, et al. Nat Immunol, 2020, 21: 727-735.). Then, the t-student test is suitable for the statistical analysis in the experiments. In addition, three samples are sufficient for the statistical analysis with t-test.

Additional comment.

Authors should perform five experiments.

Answer: Thanks for the comments. Identical number of replicates had been reported in different research articles (e.g. Zhong Zhenyu, et al. Nature, 2018, 560: 198-203.; Chen Jueqi, et al. Nature, 2018, 564: 71-76.; Jia Mutian, et al. Nat Immunol, 2020, 21: 727-735). In our paper, the experiments were conducted three times with highly reproducible. We consider this to be a sufficient sample size for demonstrating reproducibility of our findings. Thanks again for your precious suggestions.

Reviewer #2 (Remarks to the Author):

I would like to thank the authors for addressing the suggested comments. There are a couple of things that I would like to add.

Answer: We appreciate very much for your time in reviewing our manuscript. In accordance with your insightful and valuable suggestions and comments, we carefully revised the manuscript. The point-by-point answers to the comments and suggestions were listed as below.

1. I think the Immunofluorescence in figure 1 data does not add muc. In the examples you

show from previous publications sometimes NLRP3 is seen in the nucleus and sometimes not. As this figure does not really add much to your discoveries. I would particularly remove it and re-arrange the figure in the order suggested by the other reviewer.

Answer: Thank you very much for your comments. As reviewer suggested, we have deleted the immunofluorescence in Figure 1c and re-arranged the figure.

2. In line 115 "...Trim28 knockdown also reduced IL-1 β secretion following LPS priming and ATP activation in mouse" By knockdown you mean siRNA? Maybe just clarify that.

Answer: Yes. For *Trim28* knockdown experiments, we used siRNA. To avoid misunderstanding, we also clarified it in the manuscript (see line 100-102).

Otherwise I am happy with the manuscript.